# Projected long-term effects of colorectal cancer screening disruptions following the COVID-19 pandemic

Pedro Nascimento de Lima[1]\*, Rosita van den Puttelaar[2], Anne I Hahn[3], Matthias Harlass[2], Nicholson Collier[4], Jonathan Ozik[4], Ann G Zauber[3], Iris Lansdorp-Vogelaar[2], Carolyn M Rutter[5]

[1]RAND Corporation, Santa Monica, United States; [2]Erasmus MC, Rotterdam, Netherlands; [3]Memorial Sloan Kettering Cancer Center, New York, United States; [4]Argonne National Laboratory, Lemont, United States; [5]Fred Hutchinson Cancer Center, Seattle, WA, United States

**Abstract** The aftermath of the initial phase of the COVID-19 pandemic may contribute to the widening of disparities in colorectal cancer (CRC) outcomes due to differential disruptions to CRC screening. This comparative microsimulation analysis uses two CISNET CRC models to simulate the impact of ongoing screening disruptions induced by the COVID-19 pandemic on long-term CRC outcomes. We evaluate three channels through which screening was disrupted: delays in screening, regimen switching, and screening discontinuation. The impact of these disruptions on long-term CRC outcomes was measured by the number of life-years lost due to CRC screening disruptions compared to a scenario without any disruptions. While short-term delays in screening of 3–18 months are predicted to result in minor life-years loss, discontinuing screening could result in much more significant reductions in the expected benefits of screening. These results demonstrate that unequal recovery of screening following the pandemic can widen disparities in CRC outcomes and emphasize the importance of ensuring equitable recovery to screening following the pandemic.

\*For correspondence:
plima@rand.org

Competing interest: The authors declare that no competing interests exist.

## Editor's evaluation

This important study uses two well-established colorectal cancer models to estimate the potential impact of disruptions in screening caused by the COVID-19 pandemic. By dividing the population into separate cohorts based on age and pre-pandemic screening status, the authors provide convincing evidence for the adverse impact of delays in screening, switching regimens, and screening discontinuation. The finding that discontinuation has a much greater impact on screening-associated gains in life expectancy than shorter-term delays or switching of regimens suggests that access-related barriers to screening resumption may lead to the worsening of current disparities.

## Introduction

The novel SARS-Cov-2 (COVID-19) pandemic has resulted in major health consequences across the globe. In addition to the over 1 million COVID-19 deaths in the United States (*Johns Hopkins Univerity & Medicine Coronavirus Resource Center, 2022*), the pandemic has also contributed to steep declines in cancer screening, most notably in the early phases of the pandemic due to government-mandated shutdowns of non-emergency medical services (*Gupta et al., 2020*). It is estimated that colorectal cancer (CRC) screening decreased by 85% in the United States during the early phase of the pandemic, from March through April 2020 (*London et al., 2022*). The pandemic continues to affect

CRC screening and diagnosis through staff shortages that reduce capacity at gastroenterology clinics and patient hesitancy to seek care (*Wilensky, 2022*; *Del Vecchio Blanco et al., 2020*). Despite cancer screening reopening efforts, CRC screening has not yet returned to pre-pandemic levels (*Ong, 2021*).

CRC remains the second-leading cause of cancer deaths in the United States, with approximately 153,020 new cases and 52,550 deaths estimated in the year 2023 (*Siegel et al., 2023*). There is clear evidence that screening has a major impact on reducing the burden of CRC (*Edwards et al., 2010*; *Zauber et al., 2012*) and that it is cost-effective (*Knudsen et al., 2021*; *Lansdorp-Vogelaar et al., 2011*). The current United States Preventive Task Force (USPSTF) report recommends multiple screening options, including annual fecal immunochemical tests (FIT) and colonoscopy every 10 years for average-risk individuals (*Davidson et al., 2021*). However, CRC screening uptake was of concern even before the pandemic, with CRC screening rates well below the goal of 70.5% for Healthy People 2020 and the National Colorectal Cancer Roundtable goal of 80% by 2018 (*Shapiro et al., 2021*). Low rates of CRC screening have been exacerbated by the COVID-19 pandemic, and delays in screening will result in delays in diagnosis, stage progression, and increased CRC mortality.

The pandemic may also further exacerbate existing disparities related to screening. The burden of unemployment and associated loss of access to healthcare varies across different racial and ethnic groups (*Marcondes et al., 2021*). Because of this, the pandemic may contribute to widening disparities in cancer outcomes. A recent analysis using National Health Interview Survey (NHIS) data postulated that unemployment was adversely associated with being up-to-date with screening, with only 16.7% of unemployed individuals participating in recent CRC screenings, only 48.5% of whom were up-to-date with CRC screening (*Fedewa et al., 2022*).

The objective of this study is to estimate the impact of ongoing screening and treatment disruptions induced by the COVID-19 pandemic on long-term CRC outcomes. We examine 25 scenarios that reflect different levels of pre-pandemic adherence to colonoscopy and FIT screening to assess how unequal recovery in screening may contribute to widening disparities in CRC lifetime outcomes.

## Methods

This paper uses two independently developed microsimulation models of CRC, CRC-SPIN and MISCAN-Colon, to estimate the effects of pandemic-induced disruptions in colonoscopy screening for eight pre-pandemic average-CRC risk population cohorts in the United States. CRC-SPIN and MISCAN-Colon models are part of the National Cancer Institute's CISNET consortium and describe the natural history of CRC in an unscreened population based on the adenoma-carcinoma sequence. Detailed descriptions of these models and underlying assumptions may be found elsewhere (*Knudsen et al., 2021*; *Loeve et al., 1999*; *Rutter et al., 2019*, *van Hees et al., 2014*). We consider variations on two commonly used screening strategies in the USPSTF recommendations during the onset of the pandemic in March 2020 (*Knudsen et al., 2016*): Decennial colonoscopy from age 50 to 70 and annual FIT from age 50 to 75, with diagnostic colonoscopy after a positive FIT.

### Cohorts

We simulated eight pre-pandemic population cohorts that represent average-risk individuals in the United States, defined by both cohort members' age in April 2020 and their pre-pandemic screening regimens: (i) unscreened 50-year-olds (U50), (ii) unscreened 60-year-olds (U60), (iii) colonoscopy screening-adherent 60-year-olds (C60, who received their first screening colonoscopy at age 50 but have not yet had a colonoscopy at age 60), (iv) FIT screening-adherent 60-year-olds (F60, who performed annual FIT from age 50 to 59), (v) FIT screening semi-adherent 60-year-olds (f60) – those who received biannual FIT from age 50 to 56, (vi) unscreened 70-year-olds (U70), (vii) colonoscopy screening-adherent 70-year-olds (C70, who received screening colonoscopies at age 50 and 60), and (viii) FIT screening-adherent 70-year-olds (F70, who performed annual FIT from age 50 to 69). We simulated 10 million individuals within each cohort to reduce the stochastic variability in our runs and to ensure sufficient precision in our estimates. For each cohort, we simulated three sets of post-pandemic scenarios: no disruption, delays, and no screening.

## Screening regimens under no disruption

The *no-disruption* scenarios simulate post-pandemic screening scenarios for each cohort in the counterfactual scenario where no pandemic-induced screening disruptions occurred. In no-disruption scenarios, all these cohorts would have been screened during the pandemic first lockdowns in March 2020. Cohorts with colonoscopy and FIT adherent individuals (U50, C60, F60 C70, F70) continue to follow guideline-recommended strategies strictly, with no delays. Cohorts with delayed initiation (U60, U70) begin screening late but otherwise follow guideline-recommended strategies with no delays but without any additional screening beyond the usual stopping age. Finally, for the *FIT-semi-adherent 60-year-olds* (f60), we simulate resumption of biannual FIT at age 60, continuing to age 75.

## Pandemic-induced disruptions in CRC screening

### Delays

The pandemic has been shown to affect CRC outcomes through *delays* in screening. Screening colonoscopy and FIT are assumed to be delayed for a set duration of months starting at the onset of the COVID-19 pandemic in April 2020. Short-term screening delays may have occurred for a series of reasons. First, elective procedures were postponed during the first months of the pandemic. The cancellation of elective procedures caused a sharp decline in CRC screening exams during the initial phase of the pandemic (*Gupta and Lieberman, 2020*). To represent the full spectrum of delays caused by the pandemic – either due to cancellation of elective procedures or disruption in access to healthcare – we consider three sets of delays: a 3-, 9-, or 18-month delay in screening, which we label as *short-term* delays. For each delay scenario, the delay was applied on the first post-March 2020 screening exam and carried forward to any subsequent exams.

Second, the pandemic may have caused *long-term delays* in CRC screening. While the recovery in screening rates among insured individuals was rapid, (*Choy et al., 2022*) the pandemic also caused a sharp economic recession. The uneven recovery in labor force participation has the potential to cause disparities in access to healthcare in the United States due to unemployment and discontinuation of health insurance. To examine these longer-term effects of the pandemic, we consider scenarios where screening is paused for an extended period. For the 50- and 60-year-old cohorts, we simulated scenarios where screening is discontinued until the start age of Medicare enrollment (65 years). For 70-year-olds, we consider a scenario where screening is only resumed at age 75 – 5 years after the pandemic onset.

### Screening regimen switching

The pandemic may also affect CRC behavior via screening *regimen switching* – that is, changing from a colonoscopy screening regimen to one based on FIT. There is evidence that during the pandemic some patients switched from colonoscopy to FIT (*Fedewa et al., 2022*) to reduce the need for in-person endoscopy procedures. Considering this possibility, we model scenarios where individuals who initially participated in a regimen of screening colonoscopy (C60 and C70) permanently switch from decennial colonoscopy to annual FIT screening as a boundary case. While one might expect pandemic-induced regimen switching to be temporary, permanent switching can serve as a boundary case for our analysis – that is, the effect of short-term regimen switching is expected to be lower than the effect of permanent regimen switching.

### Screening discontinuation

We also simulate scenarios where screening is completely discontinued after the pandemic onset as the most consequential boundary case scenario. While only a small (unknown) proportion of individuals will discontinue screening after the pandemic, this scenario serves as an upper bound for the worst possible disruption in CRC screening following the pandemic.

### Scenarios

Each of the scenarios simulated in this study results from the combination of a pre-pandemic population cohort, a no-disruption screening scenario that serves as a counterfactual, and one or more screening disruptions (i.e. switching to FIT screening occurred in tandem with short-term delays).

**Table 1.** Study cohorts and scenarios.

| Cohort | No-disruption counterfactual | CRC screening disruption scenario | |
|---|---|---|---|
| | | Description | Label |
| Unscreened 50-year-olds (U50) | Decennial COL from age 50 to 70 | Short-term delays of [d] months* | U50 | C[d]m |
| | | Long-term delay (COL at age 65 and 75) | U50 | C@65 |
| | Annual FIT from age 50 to 75 | Short-term delays* | U50 | F[d]m |
| Unscreened 60-year-olds (U60) | Decennial COL from age 60 to 70 | Short-term delays* | U60 | C[d]m |
| | | Long-term delay (COL at age 65 and 75) | U60 | C@65 |
| | Annual FIT from age 60 to 75 | Short-term delays* | U60 | F[d]m |
| COL-adherent 60-year-olds (C60) | Decennial COL from age 50 to 70 | Short-term delays* | C60 | C[d]m |
| | | Switch to annual FIT and short-term delays | C60 | F[d]m |
| | | Long-term delay (COL at age 65 and 75) | C60 | C@65 |
| | | Discontinue screening | C60 | U |
| FIT-adherent 60-year-olds (F60) | Annual FIT from age 50 to 75 | Short-term delays* | F60 | F[d]m |
| | | Discontinue screening | F60 | U |
| FIT-semi-adherent 60-year-olds (f60) | Biannual FIT from age 50 to 56, annual FIT from age 60 to 75 | Short-term delays* | f60 | F[d]m |
| | | Discontinue screening | f60 | U |
| Unscreened 70-year-olds (U70) | COL at age 70 | Short-term delays* | U70 | C[d]m |
| | | Long-term delay (COL at age 75) | U70 | C@75 |
| | Annual FIT from age 70 to 75 | Short-term delays | U70 | F[d]m |
| COL-adherent 70-year-olds (C70) | Decennial COL from age 50 to 70 | Short-term delays* | C70 | C[d]m |
| | | Switch to annual FIT and short-term delays | C70 | F[d]m |
| | | Long-term delay Perform COL at age 75 | C70 | C@75 |
| | | Discontinue screening | C70 | U |
| FIT-adherent 70-year-olds (F70) | Annual FIT from age 50 to 75 | Short-term delays* | F70 | F[d]m |
| | | Discontinue screening | F70 | U |

Notes: This table presents the scenarios considered in this study. Each scenario corresponds to a combination of a population cohort, indicated by their age during the first COVID-19 lockdowns (March 2020), a pre-pandemic, and a post-pandemic screening regimen. The scenarios aim to represent possible combinations of screening regimens followed in the United States. The first letter in the scenario code represents screening before the pandemic and the second letter represents screening after the pandemic.

*Delays of 3, 9, and 18 months. Letter d stands for the number of months of delays.

*Table 1* lists those combinations and the scenario labels used in this analysis. We code our scenarios as *[pre-pandemic screening cohort] | [post-pandemic disruptions]*.

## Outcomes

The primary measure used to assess the benefit of CRC screening programs is the expected lifetime life-years gained (LYG) from screening. All outcomes in this study correspond to expected value of life-years (LY) across the US population with average CRC risk. This study investigates the extent to which benefits from screening are expected to be lost due to pandemic-induced disruptions to CRC

**Table 2.** Per lesion test sensitivity and specificity.

| Test | Sensitivity* | | | | Specificity† |
|---|---|---|---|---|---|
| | Adenoma 1–5 mm | Adenoma 6–9 mm | Adenoma ≥10 mm | Preclinical cancer | |
| Colonoscopy, high sensitivity‡ | 0.75 | 0.85 | 0.95 | 0.95 | 0.86 |
| Colonoscopy, low sensitivity§ | 0.55 | 0.70 | 0.90 | 0.95 | 0.86 |
| FIT¶ | | | | | |
| MISCAN | 0.00 | 0.114 | 0.159 | 0.62565/0.886 | 0.97 |
| CRC-SPIN | 0.05 | 0.15 | 0.22 | *0.74 | 0.97 |

Notes: This table presents the assumed test characteristics. We simulated two colonoscopy sensitivity scenarios seeking to represent a range of colonoscopy sensitivity of gastroenterologists in the United States.

*Sensitivity is for lesions within reach of the scope. We assume the same test characteristics for follow-up and surveillance colonoscopy as for screening colonoscopy.

†For FIT, the lack of specificity reflects detection of bleeding from other causes. We assume other-cause bleeding is independent of adenoma status. For colonoscopy, the lack of specificity reflects detection of non-adenomatous lesions, but specificity is handled in post-processing in cost-effectiveness analyses. Since this study does not consider burden outcomes, specificity is not considered in this paper. Specificity values were obtained from *Lin et al., 2021*.

‡Baseline scenarios used in *Zauber et al., 2008*.

§In line with low-sensitivity scenarios compatible with *Rutter et al., 2022*.

¶CRC-SPIN uses per-person test sensitivity for stool-based tests that are based on the size of the most advanced lesion. To account for the likelihood that a person with multiple adenomas is more likely than a person with only one to have a positive stool test, MISCAN uses lesion-based sensitivities instead of person-based sensitivities. Lesion-based sensitivities were derived by calibrating the person-based sensitivities to the number of people having one or more small/medium/large adenomas or cancers detected by stool-based testing with diagnostic colonoscopy, divided by those having one or more small/medium/large adenomas or cancers detected by colonoscopy screening.

screening. Therefore, we calculated the total number of LY for each cohort and scenario, including the number of LY under no screening (LYNS) and the number of LY under no disruptions (LYND). LYNS is computed by simulating the cohort in the absence of CRC screening and LYND is computed by simulating the same cohort under an ideal screening scenario where no disruptions to screening happened, as defined in *Table 1*.

The key outcome estimated in this study is the expected number of LY lost (LYL) due to disruptions in screening, defined as $LYL = LYND - LY$. The hypothetical number of LY gained (LYG) from screening under no disruptions are $LYG_{no\ disruption} = LYND - LYNS$. Finally, we compute the percentage of life-years gained or lost due to disruption as $\% LY\ Lost = 100 * LYL / LYG_{no\ disruption}$. The first outcome measure (*LYL*) is an absolute measure of the loss of screening benefit due to pandemic disruptions. The percent LY lost due to disruptions indicates the share of screening benefit lost due to the pandemic. Following the previous analyses, we present all outcomes as LY per 1000 individuals or life days per person. We compute each of those outcomes separately for each model and report the range of outcomes observed across both models. In addition to LY outcomes, we present lifetime number of CRC cases over the remaining lifetime of individuals and number of CRC deaths (*Supplementary files 2 and 3*).

## Test characteristics

*Table 2* specifies sensitivity and specificity assumptions underlying colonoscopy and FIT exams evaluated in this study. Our main results present colonoscopy sensitivity following assumptions used in the analysis that informed the most recent USPSTF screening recommendations (*Zauber et al., 2008*). In addition, we simulate all screening disruption scenarios under assuming lower colonoscopy sensitivity.

## CRC surveillance

We assume that individuals with an adenoma detected undergo colonoscopic surveillance according to the Multi-Society Task Force (MSTF) guidelines. These guidelines provide intervals for surveillance based on baseline findings and findings at the first surveillance colonoscopy. We assume that the

intervals provided can be more generally expressed as the intervals based on the most recent colonoscopy ('first-most recent colonoscopy') and the colonoscopy prior to that ('second-most recent colonoscopy'). In situations where the MSTF provided a range rather than a single interval, we assumed that the shortest interval would be used in routine practice. The resulting intervals are shown in *Table 3*.

We assume that persons in whom adenoma(s) have been detected remain on surveillance until age 85, provided that no adenomas are detected at the last surveillance colonoscopy. If adenomas are detected, then surveillance continues according to the clinical findings at the last colonoscopy until the person has a colonoscopy with no adenomas detected. For example, if a person has a surveillance colonoscopy at age 83 and no adenomas are detected at this exam or the exam before this one, they would be recommended to have their next surveillance at age 93. Age 93 is after the surveillance stopping age of 85 and the exam prior to age 85 was negative, so they will not have any more surveillance colonoscopies after age 83. However, if the exam at age 83 instead detected 1–2 small adenomas, they would come back for their surveillance colonoscopy at age 90, because adenomas were detected at the exam at age 83.

## Results

Loss of life due to screening disruptions was the largest for cohorts with severe disruptions after the pandemic (*Figure 1*). Aside from not receiving any screening, the worst-case scenario for the 50-year-old cohort was to postpone screening until age 65 when they become Medicare eligible. This cohort (scenario U50 | C@65) is expected to lose 104–127 LY per 1000 individuals – a 38–42% loss in LYG compared to a no-disruption scenario where they start screening at age 50 (*Table 4*). This cohort would be 1.3–1.9 times more likely to have CRC over their lifetime (*Supplementary file 2*) and 1.6–2.0 times more likely to die with CRC (*Supplementary file 3*) compared to a cohort that started screening at age 50. Other disruption scenarios are predicted to have minor effects on this cohort. For example, 50-year-olds with colonoscopy screening delayed by 18 months (scenario U50 | C18m) are expected to experience a loss of 6–7 LY per 1000 individuals, and a 2% loss in LYG from screening compared to a no-disruption scenario.

Similarly, 60-year-olds are expected to incur a substantial reduction in the benefit of screening if screening is discontinued after the pandemic. Those who started screening at age 50 and stopped after the pandemic are expected to lose 106–124 or 92–111 LY per 1000 individuals if pursuing a colonoscopy (C60 | U) or a FIT (F60 | U) screening regimen, respectively. Those who were semi-adherent to FIT screening before the pandemic and discontinued screening (f60 | U) lose even more LY – from 143 to 149 LY per 1000 individuals, or 58–69% of the benefit of screening. Similarly, unscreened 60-year-olds who start screening at age 65 (scenario U60 | C@65) are predicted to lose 42–45 LY per 1000 individuals compared to a scenario where they would have begun screening at age 60 – a 20–22% loss in LYG from screening due to this disruption.

Switching the screening regimen from colonoscopy to FIT and short-term delays will cause only a modest reduction in the benefit of screening. For the 60-year-old cohort, switching from colonoscopy to annual FIT after the pandemic with an 18-month delay is expected to result in a loss of 9–11 LY per 1000 individuals, a 3–4% loss relative to a scenario with no change in screening regimen and no delays. Similarly, short-term delays are predicted to cause minimal decreases in the benefits of the screening program. A 3-month delay in colonoscopy screening results in a loss of 0–2 LY per 1000 individuals for the 60-year cohort (scenarios C60 | C3m), whereas a 9- or 18-month delay (C60 | C9m and C60 | C18m) is expected to result in a loss of 0–2 or 0–3 LY per 1000 individuals, respectively. The worst-case scenario of an 18-month pause starting in March 2020 (scenario C60 | C18m) resulted in a 0–1% loss of the benefit of screening.

Seventy-year-olds lose fewer LY due to screening disruptions but can still be affected by the pandemic as they are at greater risk for CRC than younger age groups. When discontinuing screening after the pandemic, 70-year-olds are expected to lose 38–87 or 29–33 LY per 1000 individuals due to the pandemic if pursuing a colonoscopy (C70 | U) or FIT (F70 | U) screening regimen, respectively. Unscreened 70-year-olds who only come back to screening at age 75 (scenario U70 | C@75) are expected to lose 49–50 LY per 1000 individuals, a 39–43% reduction in LYG relative to a scenario where they would have received colonoscopy screening at age 70.

**Table 3.** CRC surveillance intervals.

| Finding at second-most recent colonoscopy[*†] | Finding at first-most recent colonoscopy[*†] | Interval[‡] to next colonoscopy, years |
|---|---|---|
| No prior colonoscopy | Normal colonoscopy | See note below§ |
| | 1–2 adenomas <10 mm | 7 |
| | 3–4 adenomas <10 mm | 3 |
| | 10 adenomas <10 mm or any adenoma ≥10 mm | 3 |
| | >10 adenomas | 1 |
| Normal colonoscopy | Normal colonoscopy | 10 |
| | 1–2 adenomas <10 mm | 7 |
| | 3–4 adenomas <10 mm | 3 |
| | 5–10 adenomas <10 mm or any adenoma ≥10 mm | 3 |
| | >10 adenomas | 1 |
| 1–2 adenomas <10 mm | Normal colonoscopy | 10 |
| | 1–2 adenomas <10 mm | 7 |
| | 3–4 adenomas <10 mm | 3 |
| | 5–10 adenomas <10 mm or any adenoma ≥10 mm | 3 |
| | >10 adenomas | 1 |
| 3–4 adenomas <10 mm | Normal colonoscopy | 10 |
| | 1–2 adenomas <10 mm | 7 |
| | 3–4 adenomas <10 mm | 3 |
| | 5–10 adenomas <10 mm or any adenoma ≥10 mm | 3 |
| | >10 adenomas | 1 |
| 5–10 adenomas <10 mm or any adenoma ≥10 mm | Normal colonoscopy | 5 |
| | 1–2 adenomas <10 mm | 5 |
| | 3–4 adenomas <10 mm | 3 |
| | 5–10 adenomas <10 mm or any adenoma ≥10 mm | 3 |
| | >10 adenomas | 1 |
| >10 adenomas of any size | Normal colonoscopy | 5 |
| | 1–2 adenomas <10 mm | 5 |
| | 3–4 adenomas <10 mm | 3 |
| | 5–10 adenomas <10 mm or any adenoma ≥10 mm | 3 |
| | >10 adenomas | 1 |

[*]A normal colonoscopy is one in which no adenomas, SSPs (not currently simulated), or CRC is detected.

[†]This table omits the case where CRC is detected at a screening, diagnostic, or surveillance colonoscopy because the CISNET CRC models do not simulate detailed events following CRC diagnosis.

[‡]The Multi-Society Task Force provides a range for some intervals (e.g. the interval for 3–4 adenomas <10 mm is 3–5 years). In such cases, we selected the shortest intervals provided.

§A person whose first screening or diagnostic colonoscopy is normal does not enter surveillance but instead resumes screening with the original modality 10 years after the normal colonoscopy. The exception to the 10-year waiting period is when the first colonoscopy is a screening colonoscopy with an $x$-year interval, where $x>10$. In that case, the next colonoscopy is in $x$ years.

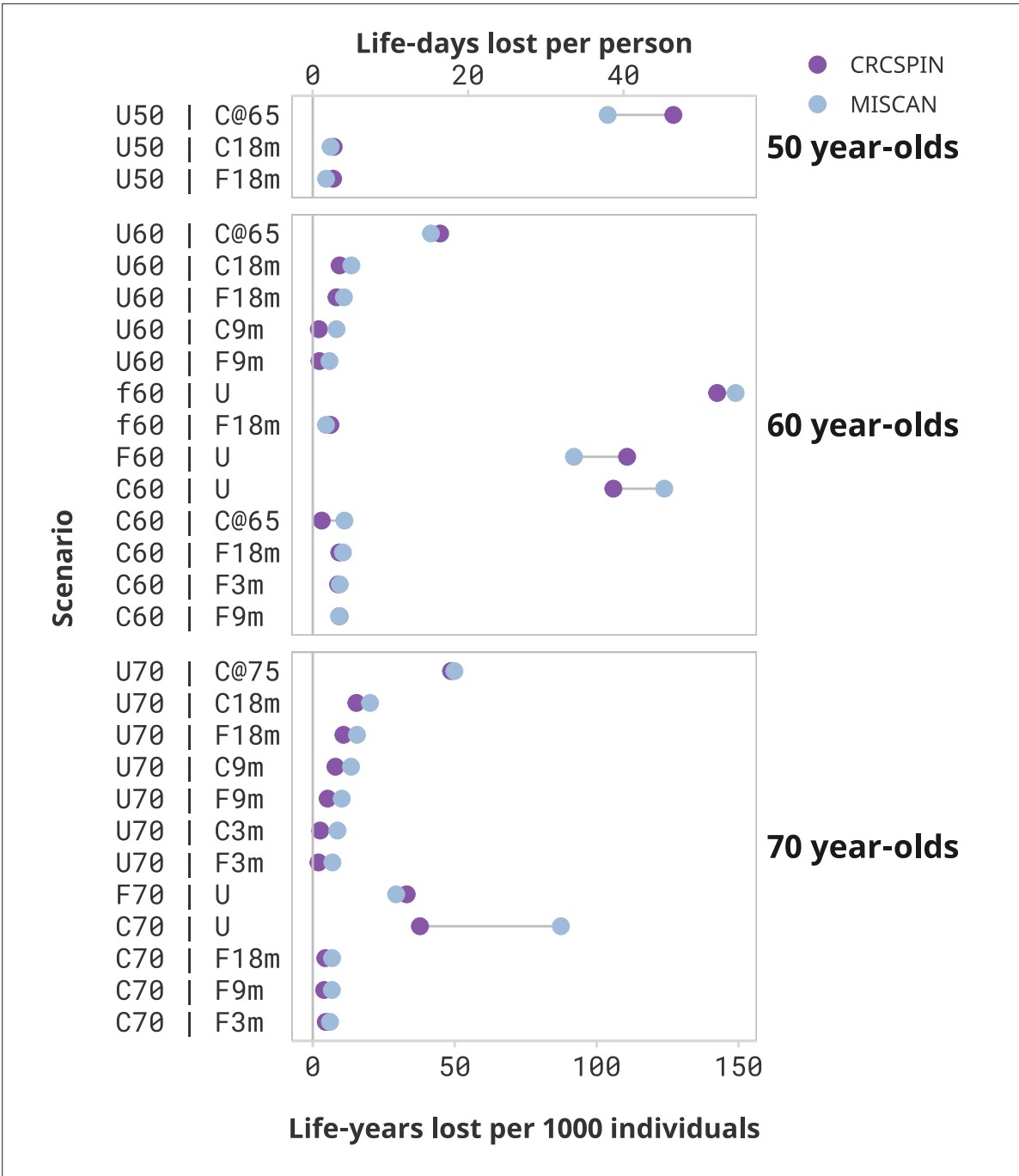

**Figure 1.** Screening benefits lost due to disruptions by cohort and scenario. Notes: Each dot represents the estimated life-years lost per 1000 individuals or life-days lost from one model under the high sensitivity scenario. Results are ordered from highest to lowest reduction in benefit induced by the pandemic. Scenarios that result in less than 2 life-days lost per person are omitted from this figure and presented in a Supplementary figure. This figure does not present a counterfactual no-screening scenario for the 50-year-olds.

The online version of this article includes the following figure supplement(s) for figure 1:

**Figure supplement 1.** Estimated loss of life in minor disruption scenarios.

Seventy-year-olds who were up-to-date with their screening and experienced short-term delays of up to 18 months can expect minimal loss of LY due to pandemic-induced CRC screening disruptions, even if they switch to FIT after the pandemic. Those who transitioned from colonoscopy to FIT screening at age 70 can expect a reduction of 5–7 LY per 1000 individuals even if a return to FIT

**Table 4.** Projected life-years (LY) per 1000 individuals.

| Scenario | Model | No screening LY [a] | Screening without disruptions LY [b] | Screening without disruptions LYG [b-a] | Screening with disruptions LY [c] | Screening with disruptions LYG [c-a] | Loss due to disruptions LY [b-c] | Loss due to disruptions % LYG loss [(b-c)/b] |
|---|---|---|---|---|---|---|---|---|
| U50 \| C3m | CRCSPIN | 31,595 | 31,893 | 299 | 31,892 | 297 | 2 | 1 |
| | MISCAN | 31,222 | 31,494 | 273 | 31,491 | 270 | 3 | 1 |
| U50 \| C9m | CRCSPIN | 31,595 | 31,893 | 299 | 31,890 | 295 | 3 | 1 |
| | MISCAN | 31,222 | 31,494 | 273 | 31,490 | 268 | 5 | 2 |
| U50 \| C18m | CRCSPIN | 31,595 | 31,893 | 299 | 31,886 | 291 | 7 | 2 |
| | MISCAN | 31,222 | 31,494 | 273 | 31,488 | 266 | 6 | 2 |
| U50 \| C@65 | CRCSPIN | 31,595 | 31,893 | 299 | 31,766 | 172 | 127 | 43 |
| | MISCAN | 31,222 | 31,494 | 273 | 31,390 | 169 | 104 | 38 |
| U50 \| F3m | CRCSPIN | 31,595 | 31,866 | 271 | 31,865 | 270 | 1 | 0 |
| | MISCAN | 31,222 | 31,483 | 261 | 31,481 | 259 | 1 | 1 |
| U50 \| F9m | CRCSPIN | 31,595 | 31,866 | 271 | 31,862 | 268 | 3 | 1 |
| | MISCAN | 31,222 | 31,483 | 261 | 31,480 | 258 | 2 | 1 |
| U50 \| F18m | CRCSPIN | 31,595 | 31,866 | 271 | 31,858 | 264 | 7 | 3 |
| | MISCAN | 31,222 | 31,483 | 261 | 31,478 | 256 | 5 | 2 |
| U60 \| C3m | CRCSPIN | 23,336 | 23,557 | 221 | 23,560 | 224 | -3 | -1 |
| | MISCAN | 23,114 | 23,309 | 195 | 23,304 | 190 | 5 | 3 |
| U60 \| C9m | CRCSPIN | 23,336 | 23,557 | 221 | 23,555 | 219 | 2 | 1 |
| | MISCAN | 23,114 | 23,309 | 195 | 23,301 | 187 | 8 | 4 |
| U60 \| C18m | CRCSPIN | 23,336 | 23,557 | 221 | 23,547 | 211 | 10 | 4 |
| | MISCAN | 23,114 | 23,309 | 195 | 23,296 | 182 | 14 | 7 |
| U60 \| C@65 | CRCSPIN | 23,336 | 23,557 | 221 | 23,512 | 176 | 45 | 20 |
| | MISCAN | 23,114 | 23,309 | 195 | 23,267 | 153 | 42 | 21 |
| U60 \| F3m | CRCSPIN | 23,336 | 23,528 | 191 | 23,529 | 193 | -2 | -1 |
| | MISCAN | 23,114 | 23,291 | 177 | 23,288 | 174 | 3 | 1 |
| U60 \| F9m | CRCSPIN | 23,336 | 23,528 | 191 | 23,525 | 189 | 2 | 1 |
| | MISCAN | 23,114 | 23,291 | 177 | 23,285 | 171 | 6 | 3 |
| U60 \| F18m | CRCSPIN | 23,336 | 23,528 | 191 | 23,519 | 183 | 8 | 4 |
| | MISCAN | 23,114 | 23,291 | 177 | 23,280 | 166 | 11 | 6 |
| C60 \| C3m | CRCSPIN | 23,243 | 23,541 | 298 | 23,541 | 299 | 0 | 0 |
| | MISCAN | 23,077 | 23,320 | 243 | 23,318 | 241 | 2 | 1 |
| C60 \| C9m | CRCSPIN | 23,242 | 23,541 | 298 | 23,541 | 299 | 0 | 0 |
| | MISCAN | 23,077 | 23,319 | 243 | 23,317 | 240 | 2 | 1 |
| C60 \| C18m | CRCSPIN | 23,242 | 23,541 | 298 | 23,540 | 298 | 0 | 0 |
| | MISCAN | 23,077 | 23,320 | 243 | 23,317 | 239 | 3 | 1 |
| C60 \| F3m | CRCSPIN | 23,242 | 23,541 | 298 | 23,532 | 289 | 9 | 3 |
| | MISCAN | 23,077 | 23,320 | 243 | 23,310 | 233 | 10 | 4 |

*Table 4 continued on next page*

*Table 4 continued*

| Scenario | Model | No screening | Screening without disruptions | | Screening with disruptions | | Loss due to disruptions | |
|---|---|---|---|---|---|---|---|---|
| | | LY [a] | LY [b] | LYG [b-a] | LY [c] | LYG [c-a] | LY [b-c] | % LYG loss [(b-c)/b] |
| C60 \| F9m | CRCSPIN | 23,243 | 23,541 | 298 | 23,532 | 289 | 9 | 3 |
| | MISCAN | 23,077 | 23,319 | 243 | 23,310 | 233 | 9 | 4 |
| C60 \| F18m | CRCSPIN | 23,243 | 23,541 | 298 | 23,532 | 289 | 9 | 3 |
| | MISCAN | 23,077 | 23,320 | 243 | 23,309 | 232 | 11 | 4 |
| C60 \| C@65 | CRCSPIN | 23,243 | 23,541 | 298 | 23,538 | 295 | 3 | 1 |
| | MISCAN | 23,077 | 23,320 | 243 | 23,308 | 231 | 11 | 5 |
| C60 \| U | CRCSPIN | 23,243 | 23,541 | 298 | 23,435 | 192 | 106 | 36 |
| | MISCAN | 23,077 | 23,320 | 243 | 23,195 | 118 | 125 | 51 |
| F60 \| F3m | CRCSPIN | 23,307 | 23,579 | 272 | 23,576 | 269 | 3 | 1 |
| | MISCAN | 23,144 | 23,377 | 234 | 23,377 | 233 | 0 | 0 |
| F60 \| F9m | CRCSPIN | 23,306 | 23,578 | 272 | 23,575 | 269 | 3 | 1 |
| | MISCAN | 23,143 | 23,376 | 234 | 23,376 | 234 | 0 | 0 |
| F60 \| F18m | CRCSPIN | 23,307 | 23,578 | 272 | 23,573 | 267 | 5 | 2 |
| | MISCAN | 23,143 | 23,377 | 234 | 23,376 | 233 | 1 | 0 |
| F60 \| U | CRCSPIN | 23,307 | 23,579 | 272 | 23,467 | 160 | 111 | 41 |
| | MISCAN | 23,143 | 23,376 | 234 | 23,283 | 141 | 93 | 40 |
| f60 \| F3m | CRCSPIN | 23,314 | 23,562 | 249 | 23,560 | 247 | 2 | 1 |
| | MISCAN | 23,136 | 23,355 | 219 | 23,353 | 217 | 2 | 1 |
| f60 \| F9m | CRCSPIN | 23,314 | 23,563 | 249 | 23,559 | 245 | 4 | 1 |
| | MISCAN | 23,136 | 23,355 | 219 | 23,352 | 216 | 3 | 1 |
| f60 \| F18m | CRCSPIN | 23,313 | 23,562 | 249 | 23,556 | 242 | 6 | 3 |
| | MISCAN | 23,136 | 23,355 | 219 | 23,350 | 214 | 5 | 2 |
| f60 \| U | CRCSPIN | 23,313 | 23,562 | 249 | 23,419 | 105 | 143 | 58 |
| | MISCAN | 23,134 | 23,353 | 219 | 23,203 | 68 | 150 | 69 |
| U70 \| C3m | CRCSPIN | 15,973 | 16,102 | 128 | 16,099 | 126 | 3 | 2 |
| | MISCAN | 15,748 | 15,866 | 117 | 15,857 | 108 | 9 | 8 |
| U70 \| C9m | CRCSPIN | 15,973 | 16,102 | 128 | 16,094 | 120 | 8 | 6 |
| | MISCAN | 15,748 | 15,866 | 117 | 15,852 | 103 | 14 | 12 |
| U70 \| C18m | CRCSPIN | 15,973 | 16,102 | 128 | 16,086 | 113 | 16 | 12 |
| | MISCAN | 15,748 | 15,866 | 117 | 15,845 | 97 | 21 | 18 |
| U70 \| C@75 | CRCSPIN | 15,973 | 16,102 | 128 | 16,052 | 79 | 50 | 39 |
| | MISCAN | 15,748 | 15,866 | 117 | 15,815 | 66 | 51 | 43 |
| U70 \| F3m | CRCSPIN | 15,973 | 16,069 | 95 | 16,067 | 93 | 2 | 2 |
| | MISCAN | 15,748 | 15,840 | 92 | 15,833 | 85 | 7 | 8 |
| U70 \| F9m | CRCSPIN | 15,973 | 16,069 | 95 | 16,063 | 90 | 5 | 6 |
| | MISCAN | 15,748 | 15,840 | 92 | 15,830 | 81 | 10 | 11 |

*Table 4 continued on next page*

*Table 4 continued*

| Scenario | Model | No screening | Screening without disruptions | | Screening with disruptions | | Loss due to disruptions | |
|---|---|---|---|---|---|---|---|---|
| | | LY [a] | LY [b] | LYG [b-a] | LY [c] | LYG [c-a] | LY [b-c] | % LYG loss [(b-c)/b] |
| U70 \| F18m | CRCSPIN | 15,973 | 16,069 | 95 | 16,058 | 84 | 11 | 12 |
| | MISCAN | 15,748 | 15,840 | 92 | 15,824 | 76 | 16 | 17 |
| C70 \| C3m | CRCSPIN | 15,683 | 15,968 | 285 | 15,968 | 285 | 0 | 0 |
| | MISCAN | 15,590 | 15,824 | 234 | 15,823 | 234 | 1 | 0 |
| C70 \| C9m | CRCSPIN | 15,683 | 15,968 | 285 | 15,968 | 285 | 0 | 0 |
| | MISCAN | 15,590 | 15,824 | 234 | 15,823 | 233 | 1 | 0 |
| C70 \| C18m | CRCSPIN | 15,684 | 15,969 | 285 | 15,968 | 285 | 0 | 0 |
| | MISCAN | 15,589 | 15,824 | 234 | 15,822 | 233 | 2 | 1 |
| C70 \| C@75 | CRCSPIN | 15,683 | 15,968 | 285 | 15,968 | 285 | 0 | 0 |
| | MISCAN | 15,590 | 15,824 | 234 | 15,819 | 229 | 5 | 2 |
| C70 \| F3m | CRCSPIN | 15,683 | 15,968 | 285 | 15,964 | 281 | 5 | 2 |
| | MISCAN | 15,590 | 15,824 | 234 | 15,818 | 228 | 6 | 3 |
| C70 \| F9m | CRCSPIN | 15,683 | 15,968 | 285 | 15,964 | 281 | 4 | 1 |
| | MISCAN | 15,590 | 15,824 | 234 | 15,817 | 228 | 7 | 3 |
| C70 \| F18m | CRCSPIN | 15,683 | 15,968 | 285 | 15,964 | 281 | 4 | 2 |
| | MISCAN | 15,589 | 15,824 | 234 | 15,817 | 227 | 7 | 3 |
| C70 \| U | CRCSPIN | 15,683 | 15,968 | 285 | 15,930 | 247 | 38 | 13 |
| | MISCAN | 15,581 | 15,815 | 234 | 15,726 | 146 | 89 | 38 |
| F70 \| F3m | CRCSPIN | 15,764 | 16,024 | 259 | 16,023 | 259 | 0 | 0 |
| | MISCAN | 15,676 | 15,902 | 226 | 15,902 | 226 | 0 | 0 |
| F70 \| F9m | CRCSPIN | 15,765 | 16,024 | 259 | 16,023 | 259 | 1 | 0 |
| | MISCAN | 15,677 | 15,903 | 226 | 15,903 | 225 | 1 | 0 |
| F70 \| F18m | CRCSPIN | 15,766 | 16,025 | 259 | 16,024 | 258 | 1 | 0 |
| | MISCAN | 15,677 | 15,903 | 226 | 15,903 | 226 | 0 | 0 |
| F70 \| U | CRCSPIN | 15,764 | 16,024 | 259 | 15,990 | 226 | 33 | 13 |
| | MISCAN | 15,677 | 15,903 | 226 | 15,873 | 196 | 30 | 13 |

Notes: Outcomes calculated over the lifetime of a cohort of 1000 average-risk, CRC-free individuals with age at pandemic defined in the scenario description. The scenario column describes colorectal cancer screening disruption scenarios, as presented in **Table 1**. Life-years (LY) and Life-years gained (LYG) are computed over the remaining lifespan of individuals starting at the beginning of 2020. All values refer to cohort-level estimates – that is, the expected LY of an average-risk person. This table presents results assuming high colonoscopy sensitivity. **Supplementary file 1** presents additional results for the low colonoscopy sensitivity scenario, and **Supplementary files 2 and 3** present CRC cases and deaths outcomes.

screening was delayed by 18 months (scenario C70 | F18m). This reduction in benefit represents a 2–3% reduction in LYG of colonoscopy-only screening.

## Low-sensitivity scenarios

While colonoscopy sensitivity affects the overall benefit of screening, *conditional on colonoscopy sensitivity*, the loss of LY due to pandemic-induced scenarios is similar across sensitivity levels. *Figure 2* compares LYG and LYL for high- and low-sensitivity scenarios. High-sensitivity scenarios are

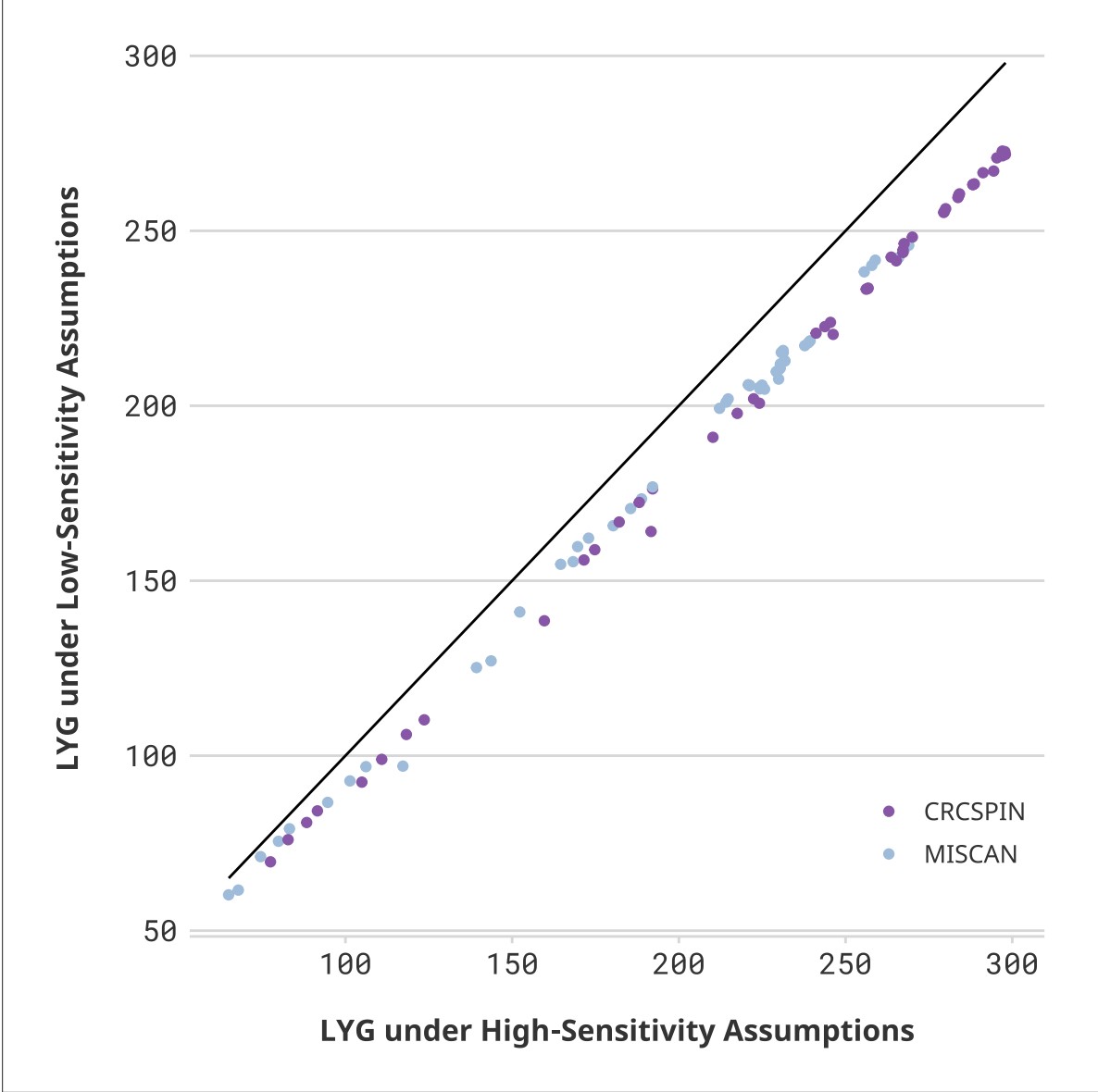

**Figure 2.** Life-years gained (LYG) in high- vs. low-sensitivity scenarios. Notes: Each dot represents one scenario considered in this study. The horizontal axis displays the number of LYG estimated in that scenario under a high colonoscopy sensitivity scenario. The vertical axis shows the results for the same cohort under a low colonoscopy sensitivity scenario. If sensitivity did not affect the estimate, then all points would be on top of a 45-degree line. Different colors represent CRCSPIN and MISCAN models.

The online version of this article includes the following figure supplement(s) for figure 2:

**Figure supplement 1.** Life-years lost in high- vs. low-sensitivity scenarios.

expected to yield higher LYG benefits than low-sensitivity scenarios, and the magnitude of this difference is higher for more intensive screening regimens. For 60-year-olds with a prior colonoscopy at age 50 who experience an 18-month delay during the pandemic (scenario C60 | C 18m), the benefit of screening is 240–297 LYG per 1000 individuals under a *high colonoscopy sensitivity* scenario, whereas it is 217–272 under a *low colonoscopy sensitivity* scenario.

Nevertheless, conditional on the sensitivity scenario, the effect of pandemic disruptions on LY lost is expected to be very similar for low-sensitivity scenarios. An 18-month delay in colonoscopy screening is expected to result in a loss of 0–3 LY per 1000 individuals for 60-year-olds assuming high sensitivity, whereas it is expected to result in a loss of 0–4 LY per 1000 individuals assuming low sensitivity.

## Discussion

Model-based screening cost-effectiveness analyses present estimates under guideline-concordant scenarios, but there are many reasons why real-world screening will not follow guidelines. Chief among them in 2020, the COVID-19 pandemic severely disrupted screening. Under those conditions, disparities in health outcomes can arise if disruptions are unevenly distributed in the population.

Our results suggest that the COVID-19 pandemic will have an uneven effect on CRC outcomes depending on whether and how fast screening is resumed after the pandemic onset. Consider three cohorts with the same pre-pandemic screening regimen and behavior: 60-year-olds with a prior colonoscopy at age 50. Cohorts that experience short-term disruptions (e.g. 3–18 months) only experience a small loss of life due to short-term delays – up to 3 LY per 1000 individuals. Those who switch from colonoscopy to FIT screening are projected to experience a greater loss of life – from 9 to 11 to LY per 1000 individuals. If screening is only resumed at age 65 (e.g. age at Medicare enrollment) or abandoned, the loss of benefits from screening could be 3–11 LY per 1000 individuals (scenario C60 | C@65). Lastly, discontinuing screening after the pandemic is projected to cause a loss of 106–124 LY per 1000 individuals, a decrease of 36–51% in the benefit of screening (scenario C60 | U). These results imply that the pandemic will become a disparity-widening mechanism if it differentially affects screening access and/or behavior across different population groups. These results also show that the pandemic is unlikely to substantially affect those whose screening is only interrupted momentarily.

These results highlight the potential implications of disruptions to preventative care due to loss of insurance following the pandemic. According to data from the Bureau of Labor Statistics Current Population Survey, more dramatic declines in the number employed during the COVID-19 pandemic were seen in Black, Asian American, and Hispanic groups (*Gemelas et al., 2022*). Moreover, data from the US Census Household Pulse Survey suggests that Black and Hispanic workers were not only more likely to be unemployed but were also more likely to be without unemployment insurance (*Mar et al., 2022*). These results provide important clinical insight on the projected impact of these populations which may guide future policy on the aftereffects of the pandemic. Those who were previously uninsured for long periods of time throughout the pandemic should resume CRC screening to mitigate the long-term effects projected in these simulations.

These results also add to the growing evidence of the implications of delayed CRC care following the COVID-19 pandemic. A microsimulation study based on a Canadian population explored scenarios of differing screening delays and transition periods due to attenuated screening volumes and found that a 6-month delay in primary screening could increase CRC incidence by 2200 cases and 960 more cancer deaths over a lifetime (*Yong et al., 2021*). A microsimulation paper based on a Chilean population illustrated similar results with respect to CRC incidence and mortality due to the screening backlog and strained patient care during the pandemic (*Ward et al., 2021*). Our results mirror these conclusions and provide new scenarios which consider the aftereffects of loss of healthcare insurance due to disparities magnified by the COVID-19 pandemic.

### Limitations

This analysis presents a series of limitations. First, we do not present population-level estimates of reductions in benefits. While doing so could prove helpful, one would have to estimate how many people will be screened following each scenario we modeled. That would require individual-level data describing the distribution of delays and screening regimen switching in the population after the pandemic, which will not be available for many years. Instead of pursuing a population-level study, we conditioned our estimates on a discrete set of pre-specified disruption scenarios. This approach makes our study feasible but prevents us from making population-level predictions. Moreover, our approach does not account for potential correlation between risk factors and disruptions – we only provide estimates using models calibrated to represent cohorts with average risk.

Second, the scenarios presented in this analysis represent only a subset of the real-world changes in screening due to the pandemic. Even in the absence of a pandemic, individuals may switch from colonoscopy to FIT, and return to colonoscopy screening. To keep this analysis tractable, we restrict the variations considered in this paper to one switch from colonoscopy to FIT. Further, we only consider changes in screening regimens immediately following the COVID-19 pandemic. Third, this analysis only considers uncertainty stemming from structural differences between models and two scenarios of test characteristics and does not evaluate parameter or sampling uncertainty. Our estimates represent

the expected value of estimates conditional on scenarios across an average-risk cohort drawn from the general US population.

Finally, this paper identifies the effect of disruptions on the effectiveness of screening interventions but does not explicitly identify policy interventions or prioritization rules to amend those inequities. Future research could use extended cost-effectiveness analysis to evaluate CRC screening interventions in the context of healthcare disparities (*Asaria et al., 2016*; *Richard et al., 2020*).

## Conclusion

This study quantified the potential effect of disruptions to colonoscopy screening and demonstrated that unequal recovery of CRC screening following the pandemic will predictably widen disparities in CRC outcomes. The COVID-19 pandemic will severely reduce the benefits of CRC screening if it causes screening discontinuation or long-term (e.g. 5 year) delays. Short-term delays of 3–18 months and regime switching from colonoscopy to FIT are not expected to have significant consequences.

## Acknowledgements

This research was supported by grant U01-CA253913 from the National Cancer Institute (NCI) as part of the Cancer Intervention and Surveillance Modeling Network (CISNET). Dr. Zauber and Ms. Hahn were supported in part by NCI CCSG P30CA008748 (PI: Vickers). This research used resources of the Argonne Leadership Computing Facility, which is a DOE Office of Science User Facility supported under Contract DE-AC02-06CH11357. We would like to thank the Argonne Leadership Computing Facility staff for their timely and critical support. This research was completed with resources provided by the Laboratory Computing Resource Center at Argonne National Laboratory. The content is solely the responsibility of the authors and does not necessarily represent the official views of the National Institutes of Health.

## Additional information

### Funding

| Funder | Grant reference number | Author |
|---|---|---|
| National Cancer Institute | U01-CA253913 | Rosita van den Puttelaar<br>Anne I Hahn<br>Matthias Harlass<br>Nicholson Collier<br>Jonathan Ozik<br>Ann G Zauber<br>Iris Lansdorp-Vogelaar<br>Carolyn M Rutter |
| NIH/NCI Cancer Center Support Grant | P30 CA008748 | Anne I Hahn |
| DOE Office of Science User Facility | DE-AC02-06CH11357 | Jonathan Ozik |

The funders had no role in study design, data collection and interpretation, or the decision to submit the work for publication.

### Author contributions

Pedro Nascimento de Lima, Conceptualization, Data curation, Software, Formal analysis, Investigation, Visualization, Methodology, Writing – original draft, Project administration, Writing – review and editing; Rosita van den Puttelaar, Conceptualization, Software, Formal analysis, Investigation, Methodology, Writing – original draft, Writing – review and editing; Anne I Hahn, Conceptualization, Investigation, Methodology, Writing – original draft, Writing – review and editing; Matthias Harlass, Software, Writing – review and editing; Nicholson Collier, Jonathan Ozik, Resources, Software, Writing – review and editing; Ann G Zauber, Funding acquisition, Methodology, Writing – review and editing; Iris Lansdorp-Vogelaar, Conceptualization, Funding acquisition, Methodology, Writing – review and

editing; Carolyn M Rutter, Conceptualization, Funding acquisition, Investigation, Methodology, Writing – review and editing

**Author ORCIDs**
Pedro Nascimento de Lima ⓘ http://orcid.org/0000-0001-9057-198X
Rosita van den Puttelaar ⓘ http://orcid.org/0000-0003-2216-6557
Anne I Hahn ⓘ http://orcid.org/0000-0003-4061-2303
Nicholson Collier ⓘ http://orcid.org/0000-0002-2376-4156
Jonathan Ozik ⓘ http://orcid.org/0000-0002-3495-6735
Ann G Zauber ⓘ http://orcid.org/0000-0002-1764-5994
Carolyn M Rutter ⓘ http://orcid.org/0000-0002-4396-8594

**Decision letter and Author response**
Decision letter https://doi.org/10.7554/eLife.85264.sa1
Author response https://doi.org/10.7554/eLife.85264.sa2

## Additional files

**Supplementary files**
• MDAR checklist
• Supplementary file 1. Projected lifetime life-years (LY) per 1,000 individuals.
• Supplementary file 2. Projected lifetime CRC cases per 1,000 individuals.
• Supplementary file 3. Projected lifetime CRC deaths per 1,000 individuals.

**Data availability**
This is a computational study based on two independently developed simulation models. Simulation output data and code used to produce the figures and Supplementary Table 3 in this paper are available at https://github.com/c-rutter/unequal-recovery-covid-19, (copy archived at swh:1:rev:0244d3219552031e056f7b5d6aa02fe03276080d). Full documentation of CISNET models used to produce the results presented in this study can be found at https://cisnet.cancer.gov/colorectal/profiles.html. Interested researchers can contact authors directly for more insight into the CISNET models.

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
