## [Editor Report]

This important study uses two well-established colorectal cancer models to estimate the potential impact of disruptions in screening caused by the COVID-19 pandemic. By dividing the population into separate cohorts based on age and pre-pandemic screening status, the authors provide convincing evidence for the adverse impact of delays in screening, switching regimens, and screening discontinuation. The finding that discontinuation has a much greater impact on screening-associated gains in life expectancy than shorter-term delays or switching of regimens suggests that access-related barriers to screening resumption may lead to the worsening of current disparities.

---

## [Decision Letter]

**Decision letter after peer review:**

Thank you for submitting your article "Unequal Recovery in Colorectal Cancer Screening Following the COVID-19 Pandemic: A Comparative Microsimulation Analysis" for consideration by *eLife*. Two peer reviewers have reviewed your article, and I have overseen the evaluation in the dual role of Reviewing Editor and Senior Editor. The following individual involved in the review of your submission has agreed to reveal their identity: Evan Myers (Reviewer #1).

Essential revisions:

As is customary in *eLife*, the reviewers have discussed their critiques with one another and with the editors. The decision was reached by consensus. What follows below is an edited compilation of the essential and ancillary points provided by reviewers in their critiques and in their interaction post-review. Please submit a revised version that addresses these concerns directly. Although we expect that you will address these comments in your response letter, we also need to see the corresponding revision clearly marked in the text of the manuscript. Some of the reviewers' comments may seem to be simple queries or challenges that do not prompt revisions to the text. Please keep in mind, however, that readers may have the same perspective as the reviewers. Therefore, it is essential that you amend or expand the text to clarify the narrative accordingly.

*Reviewer #1 (Recommendations for the authors):*

A clearly presented paper with compelling and intuitive (this is a good thing!) findings. I have only a few questions/suggestions:

1) LYG is clearly the appropriate primary outcome, but it would be of interest (and helpful to some policymakers) to have estimates of net/% changes in cancer cases and deaths.

2) Lines 304-311: I agree with this, but wouldn't it be possible to at least provide bounds based on the current age structure? Even crude weighted averages would be potentially compelling.

3) Lines 289-291: How would this be implemented? Would Medicare prioritize 65-year-olds with no previous screening? I agree that this is obviously the priority population, but how it might work is worth expanding a bit.

4) Finally, although beyond the scope of the paper, some discussion of the potential use of methods of distributional and/or extended cost-effectiveness analysis to address some of the equity issues raised here would be appropriate.

*Reviewer #2 (General comments and recommendations):*

Overall, the paper looks ok methodologically, and it addresses an important issue. The underlying models are not well described, but they have been published elsewhere and I assume extensively validated by being part of CISNET. The main shortcoming is that the effect of the pandemic is purely based on assumptions, rather than data – it would have been better if they had collected some information about how long patients put off screening during the pandemic (since it has been a few years already, this should not be difficult). But seeing that they've used a range of different assumptions, their approach seems acceptable.

The authors assume that short-term delays are a fixed duration for all patients, but in practice, this might be variable – e.g. perhaps patients with risk factors might be prioritised, or perhaps those with some clinical symptoms? Or alternatively, patients with good healthcare access (who might also be at lower risk) might have shorter durations. This is an area where collecting some data might be helpful. But again, this is a limitation rather than a fatal flaw – I think the paper as it stands is a valuable contribution.

*Essential points from the post-review interaction between reviewers:*

Rev 1: I agree with Rev 2's thoughts, which I think align with mine. The issue of prioritization that Rev 2 raises really needs to be discussed more--it's not at all clear to me how that would work in most US settings. It was a basic issue DURING the pandemic for things like elective surgery, and how systems could be put in place to equitably and efficiently handle it is a challenge. The other thing that would seem to be relatively straightforward to do would be to put some ranges around the potential impact given the current population age distribution--just take the model results for the different cohorts under different scenarios and estimate a weighted average given the age distribution of the US population in 2019-2020.

Rev 2: Agree with Rev 1's overall appraisal (good paper with some limitations). Also, Rev 1's point about prioritisation is an important one, both (i) the descriptive question (what kind of prioritisation did happen during the pandemic?) and (ii) the normative question (what kind of prioritisation SHOULD be put in place, to achieve efficient and equitable outcomes with reduced resources?) While the authors don't have the data to answer it precisely, it is an issue that deserves a bit more attention/discussion.

---

## [Author Response]

Reviewer #1 (Recommendations for the authors):A clearly presented paper with compelling and intuitive (this is a good thing!) findings. I have only a few questions/suggestions:1) LYG is clearly the appropriate primary outcome, but it would be of interest (and helpful to some policymakers) to have estimates of net/% changes in cancer cases and deaths.

We thank the reviewer for this suggestion. We now present both suggested outcomes (absolute and relative CRC cases and deaths) as Supplementary files and include these results in our discussion. As a relative outcome, now we present model-projected Risk Ratios (RRs) for CRC cases and deaths relative to a no-disruption scenario.

2) Lines 304-311: I agree with this, but wouldn't it be possible to at least provide bounds based on the current age structure? Even crude weighted averages would be potentially compelling.

Estimating the population level effect is challenging because one would also have to know more (or make more assumptions) about the screening history for each age group. Estimating population-level outcomes was beyond the scope of this paper as it would require either assumptions that would be difficult to defend or data that are not available. We acknowledge this limitation in the Discussion section.

3) Lines 289-291: How would this be implemented? Would Medicare prioritize 65-year-olds with no previous screening? I agree that this is obviously the priority population, but how it might work is worth expanding a bit.

Healthcare prioritization is a thorny ethical trolley problem, and providing definitive prescriptive advice on this issue is beyond the scope of this paper; it would be infeasible to do justice to this topic in only a few sentences or even a few paragraphs. We have now rephrased that sentence to remove the previous normative connotation, and simply state that those that were previously uninsured should resume CRC screening to avoid the negative long-term effects projected in our study.

4) Finally, although beyond the scope of the paper, some discussion of the potential use of methods of distributional and/or extended cost-effectiveness analysis to address some of the equity issues raised here would be appropriate.

Thank you for this suggestion; we now mention this topic at the end of our discussion.

Reviewer #2 (General comments and recommendations):Overall, the paper looks ok methodologically, and it addresses an important issue. The underlying models are not well described, but they have been published elsewhere and I assume extensively validated by being part of CISNET. The main shortcoming is that the effect of the pandemic is purely based on assumptions, rather than data – it would have been better if they had collected some information about how long patients put off screening during the pandemic (since it has been a few years already, this should not be difficult). But seeing that they've used a range of different assumptions, their approach seems acceptable.

We agree with the reviewer that using individual-level data to characterize the distribution of delays and other CRC screening disruptions would have been ideal, but that was outside of the scope of this paper.

The authors assume that short-term delays are a fixed duration for all patients, but in practice, this might be variable – e.g. perhaps patients with risk factors might be prioritised, or perhaps those with some clinical symptoms? Or alternatively, patients with good healthcare access (who might also be at lower risk) might have shorter durations. This is an area where collecting some data might be helpful. But again, this is a limitation rather than a fatal flaw – I think the paper as it stands is a valuable contribution.

The reviewer poses interesting questions on the potential correlation between risk, access, and delays that one might be able to address using microdata. For instance, a healthcare provider with access to individual-level data might be able to perform this analysis and investigate the hypothesis that delays were distributed differentially depending on risk factors. We now specifically note this as a limitation of our paper.

Essential points from the post-review interaction between reviewers:Rev 1: I agree with Rev 2's thoughts, which I think align with mine. The issue of prioritization that Rev 2 raises really needs to be discussed more--it's not at all clear to me how that would work in most US settings. It was a basic issue DURING the pandemic for things like elective surgery, and how systems could be put in place to equitably and efficiently handle it is a challenge. The other thing that would seem to be relatively straightforward to do would be to put some ranges around the potential impact given the current population age distribution--just take the model results for the different cohorts under different scenarios and estimate a weighted average given the age distribution of the US population in 2019-2020.

As discussed above, estimating the population level effect is challenging because one would also have to know more (or make more assumptions) about the screening history for each age group. Simply taking a weighted average given the age distribution of the US population and applying it to our estimates would not result in a meaningful population-level estimate because:

– We only estimate populations that are exactly 50, 60, and 70 years old at the onset of the pandemic; and

– We only simulate a discrete, small set amongst the much larger set of real-world screening regimens.

We believe that the reasons above are sufficient to dissuade us from trying to produce population-level estimates that would lack in credibility.

As discussed before, we also added this as a limitation to our study (see comment 2, Reviewer 1)

Rev 2: Agree with Rev 1's overall appraisal (good paper with some limitations). Also, Rev 1's point about prioritisation is an important one, both (i) the descriptive question (what kind of prioritisation did happen during the pandemic?) and (ii) the normative question (what kind of prioritisation SHOULD be put in place, to achieve efficient and equitable outcomes with reduced resources?) While the authors don't have the data to answer it precisely, it is an issue that deserves a bit more attention/discussion.

We expanded the last paragraph of the Discussion section and added the issue of prescribing prioritization rules as a limitation of our paper. As discussed in our response to Reviewer 1, doing justice to the nuances of ethics and healthcare prioritization is beyond the scope of this paper. Moreover, we do not see a reason to dwell on that topic in this paper since we just demonstrated that short-term delays (i.e., interruptions to screening during the height of the pandemic when access to healthcare is interrupted) has little impact on lifetime effects. The major contributor to widening disparities is unequal access to healthcare for other reasons, including a long job loss or lack of insurance – not lack of healthcare capacity. We now make that point in the final paragraph of our discussion.